

# Design of a Majorana trijunction

Juan Daniel Torres Luna[1*], Sathish R. Kuppuswamy[2] and Anton R. Akhmerov[2†]

**1** QuTech, Delft University of Technology, Delft 2600 GA, The Netherlands
**2** Kavli Institute of Nanoscience, Delft University of Technology,
2600 GA Delft, The Netherlands

⋆ jd.torres1595@gmail.com , † trijunction@antonakhmerov.org

## Abstract

**Braiding of Majorana states demonstrates their non-Abelian exchange statistics. One implementation of braiding requires control of the pairwise couplings between all Majorana states in a trijunction device. To have adiabaticity, a trijunction device requires the desired pair coupling to be sufficiently large and the undesired couplings to vanish. In this work, we design and simulate a trijunction device in a two-dimensional electron gas with a focus on the normal region that connects three Majorana states. We use an optimisation approach to find the operational regime of the device in a multi-dimensional voltage space. Using the optimization results, we simulate a braiding experiment by adiabatically coupling different pairs of Majorana states without closing the topological gap. We then evaluate the feasibility of braiding in a trijunction device for different shapes and disorder strengths.**

See also: Online presentation recording



## 1 Introduction

A pair of well-separated Majorana states encode the occupation of a single fermionic state non-locally as two zero-energy states [1]. Under the exchange of two Majorana states—braiding—the protected ground state evolves via unitary operations. The discrete nature of braiding allows implementation of all Clifford operations with very low error rates—a requirement for universal fault-tolerant quantum computation [2]. This has brought a lot of attention to the field in the past two decades with several proposals for experimental realization [3, 4] and detection [5–7] of Majorana bound states. Therefore, there are several proposals for braiding that include moving Majoranas around each other in semiconductor nanowire networks [8, 9], long-range coupling of Majorana islands connected by quantum dots [10–13], and networks of Josephson junctions connected by trijunctions [14, 15].

Braiding in hybrid semiconductor-superconductor devices requires coupling all Majorana states via control of the electrostatic potential. Two-dimensional electron gases (2DEGs) are suitable for realizing trijunction devices because they combine different ingredients such as

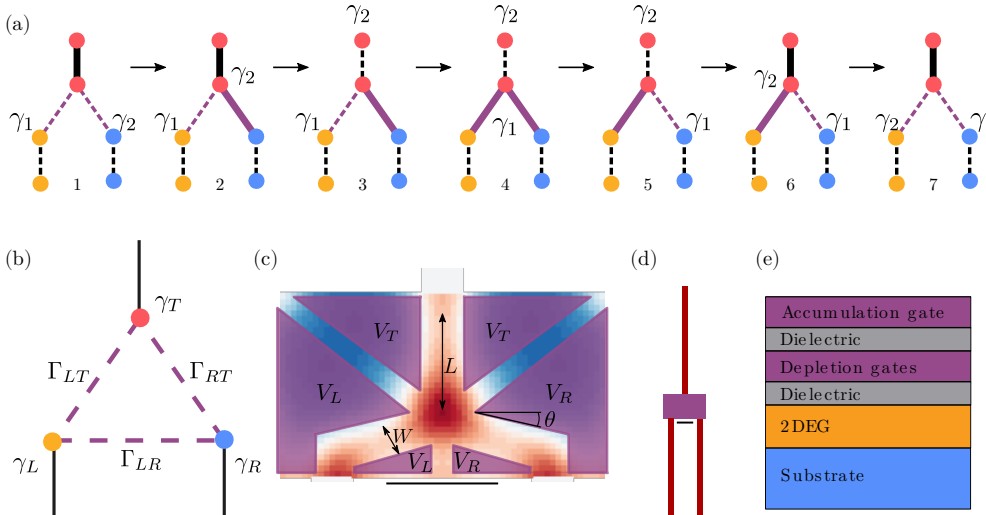

Figure 1: (a) The braiding protocol that we consider [15]. The lines indicate Majorana couplings that are either on (thick lines) or off (dashed lines). (b) A schematic of a trijunction device showing three Majorana states closest to the trijunction region (red, blue, and yellow) and their pairwise couplings. (c) Real space trijunction layout. The shape of the depletion gates (purple) is parametrized by $L$, $W$, and $\theta$. The background color shows the chemical potential. Blue regions are depleted and red regions are not. The scale bar is 200 nm. (d) The complete simulated device. The trijunction is in the middle (purple rectangle), with the nanowires (red lines) attached to it. (e) Heterostructure configuration.

electrostatic control and superconductivity [4] in a non-linear layout. 2DEGs are an active field of research for topological physics with experiments focused on detecting signatures of Majorana states in single nanowires [16–18], planar Josephson junctions [19, 20], or in minimal realizations of the Kitaev chain [21, 22]. Unambiguous detection of Majoranas requires distinguishing them from non-Majorana physics producing similar results [23, 24]. The recently proposed topological gap protocol [7] establishes a first step towards fully automated detection of Majorana states.

A braiding experiment poses additional requirements to the creation of spatially isolated Majoranas. It requires measurement of the fermion parity of Majoranas belonging to the same nanowire [25, 26]. Furthermore, it also requires a trijunction—a switch that selectively couples Majoranas from three different nanowires—which is the focus of our work. The requirements for a braiding experiment are such that (i) the energy of the coupled pairs needs to be larger than the thermal broadening, (ii) the ratio of the energies of coupled pairs with the remaining Majoranas should be as large as possible to ensure adiabaticity, and (iii) the gap between the zero-energy ground state and the coupled Majoranas does not close while coupling different pairs. A trijunction device that satisfies these requirements is suitable to perform braiding.

To evaluate the feasibility of a braiding experiment, we design and simulate a trijunction device as shown in Fig. 1. To find the operational regime of the device, we use an optimization approach using an effective Hamiltonian in the basis of decoupled Majorana states. Then, we illustrate the device operation by simulating the braiding protocol from Ref. [15] where we switch the coupling between different pairs of Majorana states while preserving the energy gap. We define quality metrics relevant for braiding and systematically compare the performance of different trijunction device geometries. We highlight the geometries that are suitable for braiding and investigate their resilience to increasing concentration of electrostatic disorder that is unavoidable in this system [7].

## 2 Device layout and braiding protocol

A braiding protocol [15, 27] requires time-dependent manipulation of the pair couplings between three Majorana states shown in Fig. 1(a). The computational subspace—one Majorana in the trijunction and three Majoranas in the far nanowires' ends—is protected as long as the number of zero-energy modes remains constant. In other words, the computation is protected as long as two out of six Majorana states are always coupled. The full braiding protocol requires coupling Majoranas from the same wire via a transmon [26] or flux qubit [25], which is outside the scope of this work. It also requires moving one Majorana state between three different wires by coupling different pairs of Majoranas via a trijunction. By combining these two procedures, it is possible to perform a braiding experiment where two Majorana states exchange positions.

We adapt the braiding protocol from Ref. [15] that exchanges Majoranas $\gamma_1$ and $\gamma_2$ as shown in Fig. 1(a). The ingredients that we require for the braiding protocol are

- coupling Majoranas within the same nanowire via charging energy [25, 26],

- coupling pairs of Majoranas via the trijunction,

- coupling all three Majoranas in the trijunction as in step 5 of Fig. 1(a),

- a path in parameter space that interpolates between a regime with two Majoranas coupled to the regime with three Majoranas coupled without closing the topological gap, that is, a path with a finite gap during steps 3, 4, and 5 of Fig. 1(a).

Our goal is to compute the coupling of different Majoranas required to implement the braiding protocol of Fig. 1(a). Because the purpose of our study is the design of the trijunction, we exclusively consider the three Majoranas closest to the trijunction which interact via the potential in the middle region shown in Fig. 1(c). Therefore, we do not consider the Coulomb couplings between the Majoranas in the nanowires shown in steps 1 and 7 in Fig. 1(a). For the same reason, we leave to future work the analysis of the competition between Coulomb-mediated Majorana coupling and the direct coupling at the trijunction [28]. Furthermore, because the on/off ratios of the couplings are sufficient to determine whether braiding can be performed adiabatically, we do not simulate the explicit time dependence of gate voltages. Finally, detailed modeling of Majorana nanowires is outside the scope of our study. Therefore, we consider an idealized model of topological nanowires.

We simulate clean nanowires of size $W_{NW} = 70\,\text{nm}$ and $L_{NW} = 1.5\,\mu\text{m}$ such that the Majoranas are well-separated. An external magnetic field is parallel to the nanowires and drives them into the topological phase. We connect the nanowires to the trijunction formed in the central normal region as shown in Fig. 1(c). We use one layer of depletion gates shown in Fig. 1(c) to form the trijunction and a second layer for a global accumulation gate to control the electron density. We parameterize the shape of the device using channel length $L$, channel width $W$, and the angle $\theta$ between the $x$-axis and the arms. We use the materials from Ref. [29] for the substrate, dielectric, and gate electrodes.

We simulate the three-dimensional device configuration shown in Fig. 1(c-e). We use the electrostatic solver of Ref. [30] to numerically solve the Poisson's equation

$$\nabla \cdot [\epsilon_r(\mathbf{r})\nabla U(\mathbf{r})] = -\frac{\rho(\mathbf{r})}{\epsilon_0}, \tag{1}$$

where $\rho_r$ is the charge density, $\epsilon_0$ is the vacuum permittivity and $\epsilon_r$ is the relative permittivity. Because the 2DEG has a low electron density, we neglect the potential induced by charges in

the 2DEG. We express $U$ as a linear combination of the potential induced by each gate electrode

$$U(\mathbf{r}) = \sum_i V_i U_i(\mathbf{r}) + U_0(\mathbf{r}), \qquad (2)$$

where $U_0(\mathbf{r})$ is the potential induced by dielectric impurities when $\mathbf{V} = 0$, and $V_i$ are the elements of $\mathbf{V} = (V_L, V_R, V_T, V_{\text{global}})$. To reduce the number of control parameters, we apply the same voltages to the depletion gates closest to a channel shown in Fig. 1(c).

We use the 2D Hamiltonian

$$H = \left( \frac{1}{2m^*}(\partial_x^2 + \partial_y^2) - U(x, y) \right)\sigma_0 \tau_z + \alpha(\partial_x \sigma_y - \partial_y \sigma_x)\tau_z + E_z \sigma_y \tau_0 + \Delta(x, y)\sigma_0 \tau_x, \quad (3)$$

where $\sigma_i$ and $\tau_i$ are the Pauli matrices in the spin and particle-hole space, $\alpha$ is the spin-orbit coupling strength, $E_z$ is the Zeeman field induced by the homogeneous magnetic field, and $m^*$ is the effective mass in the semiconductor. Using the Kwant software package [31], we discretize Eq. (3) over a 2D tight-binding square lattice with lattice constant $a =$10nm as for typical devices [32]. The electrostatic potential in the 2DEG, $U(x, y, z = 0) = U(x, y)$, is defined relative to the Fermi level in the nanowires which is set to the bottom of the lowest transverse band $\mu$. The superconducting pairing is absent in the normal region, and in the nanowires, it is $\Delta(x, y) = \Delta_0 e^{i\phi_j}$ where $\Delta_0$ is the induced gap and $\phi_j$ is the phase in the $j$-th nanowire. We tune the Hamiltonian to be in the topological phase for the lowest subband, $E_z > \sqrt{\mu^2 + \Delta_0^2}$. The topological gap in the nanowires is $\Delta_t$. The parameters used in the Hamiltonian and the electrostatic simulation are listed in Appendix A.

## 3 Device tuning

To determine couplings of individual Majoranas from the low-energy eigenvalue decomposition of the Hamiltonian, we need to interpret the wave functions in terms of Majoranas belonging to different wires. We do this by first considering a point in the parameter space where the trijunction is disconnected and use it to define the reference Majorana wave functions. We numerically compute the six lowest energy modes $|\phi_i\rangle$ of the full device shown in Fig. 1(d) when the normal region is depleted. The eigenstates $|\phi_i\rangle$ are linear combinations of decoupled Majorana states $|\gamma_i\rangle$. We obtain a basis of individual Majorana states $|\gamma_i\rangle = \hat{W}|\phi_i\rangle$, where $\hat{W}$ is the matrix that simultaneously approximately diagonalizes the projected position operators $\hat{\mathbf{P}}_x = \langle\phi_i|\hat{\mathbf{X}}|\phi_j\rangle$ and $\hat{\mathbf{P}}_y = \langle\phi_i|\hat{\mathbf{Y}}|\phi_j\rangle$. After Wannierization, we fix the phase of $|\gamma_i\rangle$

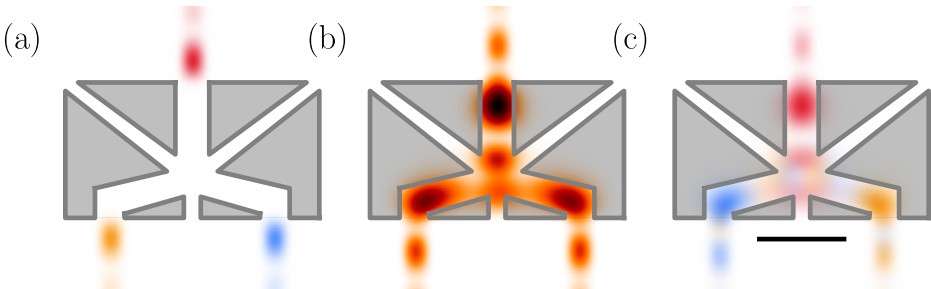

Figure 2: Representation of coupled Majoranas in the basis of localized states. (a) Densities of 3 decoupled Majorana states $|\gamma_i\rangle$. (b) wave function of coupled Majoranas $|\psi_i\rangle$. (c) Decomposition of the coupled wave function into decoupled states using the SVD.

so that $\mathcal{P}|\gamma_i\rangle = |\gamma_i\rangle$), making $|\gamma_i\rangle$ their own particle-hole partners. To determine the effective trijunction Hamiltonian, we project out the decoupled Majorana states at the far ends of the wires and keep only the Majorana states that are closest to the middle region, as shown in Fig. 2(a). When the three Majoranas are strongly coupled as in Fig. 2(b), the eigenstates $|\psi_j\rangle$ are not linear combinations of decoupled Majorana states. However, the three eigenstates closest to the trijunction form a particle-hole symmetric subspace where any fermionic state can be expressed as a linear combination of the individual Majorana states $|\gamma_i\rangle$ as in Fig. 2(c).

We interpret the low energy eigenstates localized in the trijunction $|\psi_i\rangle$ as linear combinations of Majoranas originating from different arms by computing the overlap matrix $S_{ij} = \langle \gamma_i | \psi_j \rangle$. We then apply a singular value decomposition (SVD), $S = UDV^\dagger$, where $U$ and $V$ are unitary and $D$ is positive diagonal. The approximate transformation is the unitary part of the SVD, $S' = UV^\dagger$. This transformation corresponds to choosing the coupled Majorana wave functions as $|\gamma'_j\rangle = \sum_{jk} S'_{jk} |\psi_k\rangle$ as shown in Fig. 2 (b-c). The low-energy effective Hamiltonian is

$$H_{\text{eff}} = S' \text{diag}(-E_1, 0, E_1) S'^\dagger = i \sum_{i \neq j} \Gamma_{ij} |i\rangle \langle j|, \tag{4}$$

where $\Gamma_{ij} = -\Gamma_{ji}$ is the coupling between Majoranas $\gamma'_i$ and $\gamma'_j$, and $E_1$ is the energy of the first excited state of the system. When only two Majoranas are coupled, their effective coupling $|\Gamma_{ij}| = E_1$, however, when there are multiple pairs of coupled Majoranas, the interpretation of the effective couplings $\Gamma_{ij}$ is ambiguous.

## 4 Optimizing pairwise couplings

Initially, we consider steps 2, 3, 5, and 6 of Fig. 1(a) where a single pair of Majoranas is connected via the trijunction. We use an optimisation approach to find the optimal couplings as a function of gate voltages and phase differences. For the coupling of the $i$-th and $j$-th Majorana states, we define the desired and undesired couplings as

$$\delta_+ = |\Gamma_{ij}|, \quad \delta_- = |\Gamma_{ik}| + |\Gamma_{jk}|, \tag{5}$$

where $k$ is the remaining Majorana state. The goal of our device is to maximize the energy of the coupled Majorana pair while keeping the couplings to the remaining Majorana state exponentially small. Therefore, we define a loss function that maximizes the desired coupling and minimizes the undesired coupling:

$$C_{\text{pair}} = -\delta_+ + \log(\delta_-^2 + \epsilon). \tag{6}$$

Here, $\delta_\pm$ is in units of $\Delta_t$. We use $\epsilon = 10^{-3}$ to regularize the divergence of the logarithm.

To remove the local minima of the loss function and improve the convergence, we penalize the regions in the gate voltage space where either the regions under the gate are not depleted or the channels are fully depleted. We achieve this by adding the following soft-threshold terms to the loss function:

$$S(U(\mathbf{r})) = A\left(\sum_{\{\mathbf{r}_{\text{acc}}\}} U(\mathbf{r}_{\text{acc}})\Theta[U(\mathbf{r}_{\text{acc}})] + \sum_{\{\mathbf{r}_{\text{dep}}\}} (U(\mathbf{r}_{\text{dep}}) - u_0)\Theta[-U(\mathbf{r}_{\text{dep}})]\right). \tag{7}$$

Here $\Theta(x)$ is the Heaviside function. We choose $\{\mathbf{r}_{\text{acc}}\}$ and $\{\mathbf{r}_{\text{dep}}\}$ to be in the accumulated channel and in the depleted regions, respectively. We choose the scale factor $A = 10^2$, and use a threshold $u_0 \sim 1 - 2\text{meV}$. The total loss function is

$$L = C_{\text{pair}} + S. \tag{8}$$

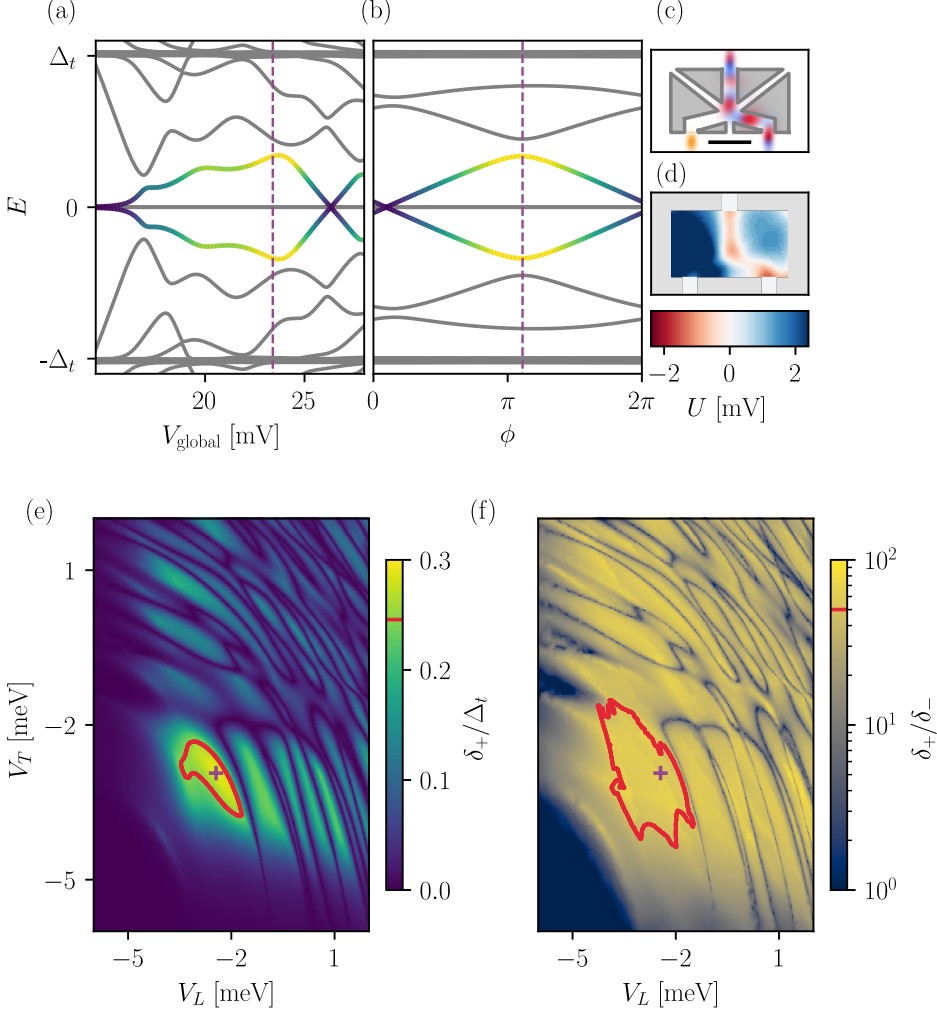

Figure 3: Spectra of a trijunction with optimally coupled $R - T$ pair of Majoranas (colored lines) with respect to (a) global accumulation gate and (b) superconducting phase difference. The optimal point is indicated by the purple dashed lines. The wave functions (c) and the potential (d) at the optimal point. Two-dimensional scans as a function of the voltages for the $T$ and $R$ depletion gates of the desired coupling (e) and the ratio of desired over undesired coupling (f). The optimal point is shown as a purple cross inside of the scan. The operation range is the area enclosed by the red line that satisfies $\delta_+ \leq 0.85 \times \delta_+^{\max}$ and $\delta_+/\delta_- > 50$.

Minimizing this loss function for all Majorana pairs yields the voltage configurations where two Majorana states are optimally coupled. The results for the $R$–$T$ pair are shown in Fig 3. At the optimal point, the depletion gates form a channel between the $R$ and $T$ Majorana states while disconnecting the $L$ Majorana as shown in Fig. 3(c-d). Once the channel is formed by the depletion gates, the coupling is controlled by tuning the accumulation gate voltage $V_{\text{global}}$ as shown in Fig. 3(a). The phase difference between the top and right superconducting arms modulates the coupling $\Gamma_{LR}$ as shown in Fig. 3(d).

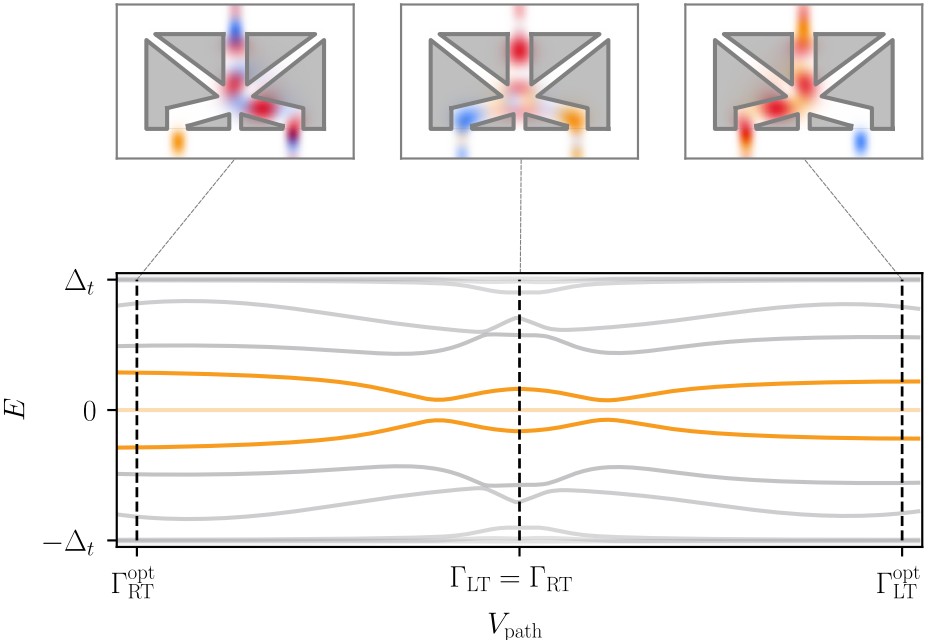

Figure 4: Exchange of two Majoranas by adiabatically coupling different pairs of Majoranas. (Top) Majorana wave functions at the optimal points where one or two pairs of Majoranas are coupled. (Bottom) The spectrum of the trijunction along the voltage path that interpolates between the optimal points.

## 5 Optimizing triple coupling

In order to couple all three Majorana states, at least two pairs of Majoranas must be coupled. Because the device without disorder is symmetric around the $x$ axis, we choose to couple the $L-T$ and $R-T$ pairs of Majoranas simultaneously, and constrain the voltages to be symmetric, i.e. $V_L = V_R$. Furthermore, since finding the optimal path in voltage space is hard, we choose the path that linearly interpolates between the point where two Majorana states are coupled and the point where all Majorana states are coupled, corresponding to steps 3, 4, and 5 of Fig. 1(a). In order to find a triple-coupled point, the loss function must maximize at least two couplings simultaneously. Furthermore, depending on the choice of the triple coupled point, the gap along the path interpolating between the pairwise coupling and the triple point may close. In the trijunction that we have studied, we find that the following loss function finds a triple coupled point connected by a gapped path to the pair of coupled points:

$$C_{\text{triple}} = -(|\Gamma_{LT}| + |\Gamma_{RT}|) + |\Gamma_{LR}|. \tag{9}$$

The gap reaches a minimum $\approx 0.1 \times \Delta_t$ along the braiding path. We obtain the optimal coupling by minimizing the loss function as in Eq. (9) plus the corresponding soft-threshold to accelerate convergence. The resulting spectrum of the trijunction has a finite gap during the entire voltage path as shown in Fig. 4. The wave functions at the optimal points are shown in the upper row of Fig. 4.

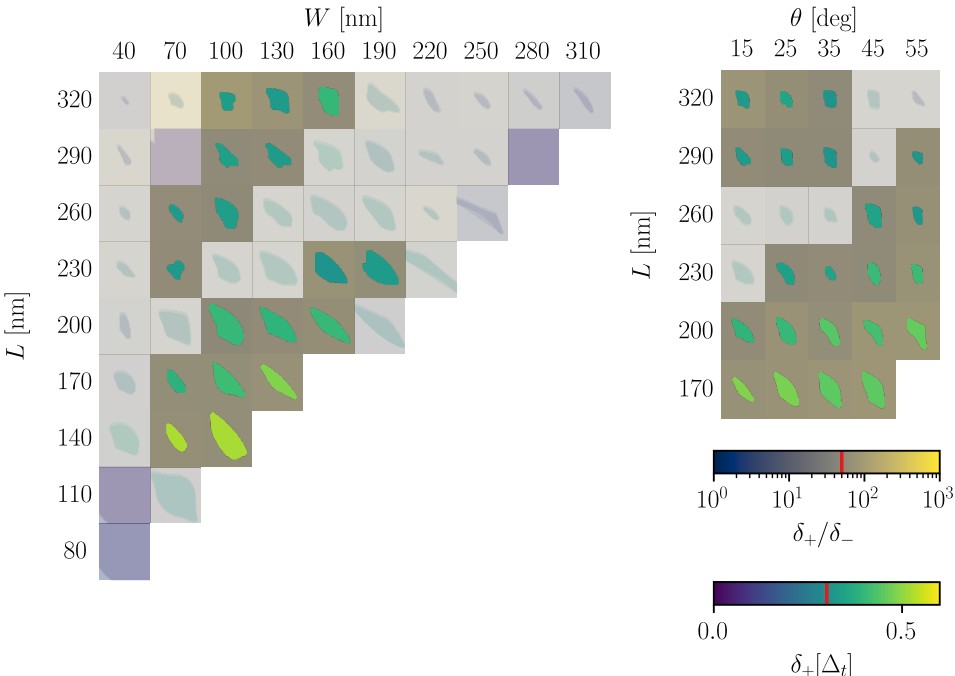

Figure 5: Analysis of quality metrics for the worst performing pair for different tri-junction geometries with $\theta = 15°$ (left) and $W = 130\,\text{nm}$ (right). The operation range for each geometry is shown inside each square and colored with $\delta_+$ at the optimal point. The background is colored with $\delta_+/\delta_-$ at the optimal point. The optimality criteria from Eqs. (10) and (11) is indicated by a red line in the respective color bar. The geometries that do not satisfy these criteria have increasing transparency. The squares fully covered in purple are the cases when the optimization algorithm did not find a solution.

## 6 Geometry dependence

In order to evaluate the adiabaticity of the braiding protocol, we compute the desired coupling, $\delta_+$, and the ratio between desired and undesired couplings, $\delta_+/\delta_-$, at the optimal point. Because the topological gap is small, we require the Majorana couplings to be comparable to it:

$$\delta_+ \gtrsim 0.3 \times \Delta_t \,. \tag{10}$$

As a minimum requirement for adiabaticity, the desired coupling should be larger than the undesired coupling:

$$\delta_+ \gtrsim 50 \times \delta_- \,. \tag{11}$$

The large ratio between desired and undesired couplings ensures that there exists a time scale $\tau$ where the device operates such that $\delta_- < \hbar/\tau < \delta_+$. Furthermore, the coupling $\delta_+$ must be larger than the thermal broadening. In order to characterize the robustness of device operation with respect to variations in the gate voltages we define the operational range $\mathcal{A}$ of the device as the area in the voltage space that satisfies both Eqs. (10) and (11). In Fig. 3(e-f) we show the operational regime of the device around the optimal point for the desired coupling and the ratio between the desired and undesired couplings, respectively. While the numerical values of the thresholds that we use are somewhat arbitrary, they leave sufficient room for adiabatic braiding while not introducing additional limitations to the device's performance.

In order to determine which geometries are suitable for braiding, we compute the quality metrics $\delta_+/\Delta_t$, $\delta_+/\delta_-$, and $\mathcal{A}$ for different $L$, $W$, and $\theta$. We evaluate the quality metrics for the worst-performing pair. We summarize the results in Fig. 5 and indicate the geometries that meet the thresholds of Eqs. (10,11). We find that the quality of a trijunction depends on the length scales of the normal region. Because Majorana couplings decay with distance, small trijunctions have a systematically larger operational voltage range. In very small trijunctions, however, it becomes impossible to suppress unwanted couplings. Furthermore, there is an optimal aspect ratio between length $L$ and width $W$ that guarantees control over the individual channels formed in the trijunction arms. The angle $\theta$ does not affect the qualitative behavior of the trijunction.

# 7 Electrostatic disorder

We compare the susceptibility to electrostatic disorder of larger and smaller geometries. For that, we select two geometries and analyze their performance in the presence of disorder. We simulate disorder in the dielectric between the depletion gate layer and 2DEG by randomly positioned positive charges. Figure 6 shows that devices with an impurity concentration of $\sim 10^{10}\,\mathrm{cm}^{-2}$ are not degraded by disorder. On the other hand, a small concentration of electrostatic disorder $\sim 10^{11}\,\mathrm{cm}^{-2}$, which is reported to be achieved in Ref. [7], significantly reduces the performance of a trijunction. While smaller geometries perform better, we expect that they are more susceptible to fabrication imperfections, therefore posing a tradeoff between two challenges.

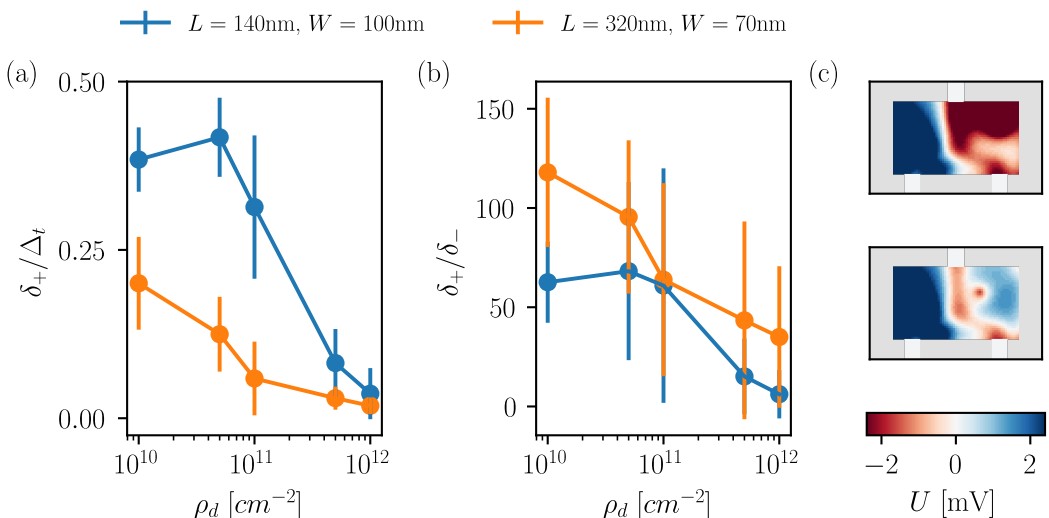

Figure 6: Impact of disorder on $\delta_+$ (a) and $\delta_+/\delta_-$ (b). We show two example disordered realizations (c) for $\rho = 10^{10}\,\mathrm{cm}^{-2}$. We considered 10 disorder realizations for each impurity density. The error bars correspond to the standard deviation.

# 8 Summary

In this work, we developed a numerical procedure to design a braiding protocol using a trijunction device—one of the ingredients for a topologically protected quantum computer—by using three-dimensional electrostatic and quantum simulations. We used an optimization approach to find the voltage configurations where all different pairs of Majorana states are strongly coupled. Consequently, we discovered that a range of trijunction device geometries can be used as switches that selectively couple and decouple different Majorana states. We confirmed that trijunctions are suitable for braiding by simulating the braiding protocol from Ref. [15] without closing the gap between the ground state and the coupled Majorana states. The operation of the device is limited by the gap size, which decreases to $\lesssim 0.1 \times \Delta_t$ along the braiding protocol. We observe that state-of-the-art levels of disorder render this trijunction design inoperable because the narrow channels cannot be formed. Therefore, we expect that cleaner materials [33] or a different design would be required to resolve this problem.

The methods developed in our study apply to other realizations of Majorana states such as the minimal Kitaev chain [21,34]. Similarly, the optimization method that we developed is transferable to other semiconducting devices such as spin qubits [35] or hybrid devices such as planar Josephson junctions [32]. The operational regime of these devices usually lies in a region of a multidimensional space that maximizes certain quantities such as the wave function overlap [35] or the energy gap [32]. Our work demonstrates that combining electrostatic simulations, effective Hamiltonians, and optimization routines is a powerful tool in designing and operating semiconductor devices.

# Acknowledgments

We thank C. Liu, V. Fatemi, H. Spring, J. Zijderveld, K. Vilkelis, C. Prosko, C. Moehle, and S. Goswami for useful discussions. We thank I. Araya Day for help with the algorithms for identifying the effective Hamiltonian.

**Author contributions**   A.R.A. defined the project goal and supervised the project. J.D.T.L. designed the trijunction device. J.D.T.L. and S.R.K. set up the simulations and obtained the results. J.D.T.L. wrote the manuscript with input from S.R.K. and A.R.A.

**Data availability**   All code and data used in this work are available at Ref. [36].

**Funding information**   This work was supported by the Netherlands Organization for Scientific Research (NWO/OCW) as part of the Frontiers of Nanoscience program, an ERC Starting Grant 638760, a subsidy for top consortia for knowledge and innovation (TKI toeslag), and a NWO VIDI Grant (016.Vidi.189.180).

# A   Simulation details

The values used in Eq.(3) are $t = \hbar^2/2m^*$ where $m^* = 0.023 \times m_e$ and $m_e$ is the electron mass, the bare superconducting gap is $\Delta_0 = 0.5\,\mathrm{meV}$, the spin-orbit interaction $\alpha = 3 \times 10^{-11}\,\mathrm{eV\,m}$, the Zeeman field that drives the nanowires in the topological phase is $E_Z = 1.0\,\mathrm{meV}$, and the nanowire chemical potential at the bottom of the lowest band is $\mu = 2.396\,\mathrm{meV}$. The topological gap is $\Delta_t = 0.325\,\mathrm{meV}$. The coherence length in the nanowires is $\xi_{sc} \approx 80.2\,\mathrm{nm}$

and the localization length of the Majoranas is $\xi_{\text{MZM}} \approx 487.474$ nm. Similarly, in Table 1 we detail the parameters used to set up the three-dimensional electrostatic simulation and solve Eq.(1).

Table 1: Parameters used in the electrostatic simulations. The heterostructure layers are shown in Fig.1(e) and their corresponding thicknesses and relative dielectric permittivities with respect to the vacuum permittivity $\epsilon_0$ are detailed here. The metallic gates have infinite permittivity and a thickness of 30nm.

| Layer | Thickness [nm] | Relative permittivity |
|---|---|---|
| Substrate | 50 | 16 |
| 2DEG | 20 | 15 |
| Dielectric | 30 | 9.1 |

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
