# Peer review of "Design of a Majorana trijunction"

_SciPost Physics, doi:SciPost Phys. 16, 044 (2024)_

## Round 2 · List of Changes

# Reply to referees

## Reply to the Referee 1

We thank the referee for reading our manuscript and commenting on possible points of improvement. In the updated version we have addressed all points raised by the referee.

> (1) The authors have a confusing mix of physical and absolute units when defining the Hamiltonian/setup in Eq. (1-3). I think it would be useful if the authors had a table, or equivalent, outlining all the physical parameters they utilise in their simulations. (I apologise if I have missed these listed somewhere in the text, but even if they are somewhere in the text I still feel it would be useful to have them centralised since I cannot find them).

In order to make it easier to find all the parameters that we used in the simulations, we added a corresponding appendix.

> The value of the dielectric constant is not listed. It would also be useful to discuss what influence this will have on e.g. the disorder discussion at the end.

We have added the value of the dielectric constant to the parameter listing in the appendix. At this point we do not have concrete conclusions regarding optimal material choices and believe that a proper analysis extends beyond comparing the dielectric constants.

> In the final lines of section 2 there are 3 pairing potentials: $\Delta_0$, $\Delta$, and $\Delta_t$. From the description it appears that these are all (probably) the same. If they are not then it should be made clear how they are related.

The different Delta's are detailed in the last few sentences of the last paragraph of Sec. 2.

> (2) Is there any firmer justification for the thresholds outlined in Eqs. (10-11)? It would be useful to understand what these thresholds (roughly) mean in terms of operational temperatures/timescales. At the moment they just appear to be rather arbitrarily chosen.

In the new version of the manuscript we explain the reasoning behind Eqs. (10, 11) and why these requirements must be satisfied to achieve braiding. We agree that the values of the thresholds are somewhat arbitrary. These values roughly correspond to minimal requirements for braiding. We now state this in the updated manuscript.

> (3) As stated above, Fig. 5 would be greatly improved by understanding the size of the coherence length in comparison with the junction geometry. However, I also do not really understand the physics behind the statement “smaller geometries perform better”. Is there any way of understanding this finding? At first sight it is rather surprising given that one would expect unwanted couplings to be larger in smaller junctions.

We have explained in the manuscript that the performance of the trijunction depends on the length scales of the normal region rather than on the length scales of the superconductor. We have added the coherence length and the Majorana localization length to the appendix for completeness.

> (I am also not sure I agree that ~ $10^{11}$ cm$^{-2}$ is really achieved in Microsoft’s experiment).

We have reworded the citation to use a more neutral stance on the Microsoft claim ("reported" instead of "achieved").

## Reply to Referee 2

We thank the referee for reading our manuscript and raising points for improvement. In the updated version we address all points raised by the referee.

> (1) While it may be obvious to the authors, I think they should elaborate on the set-up of a "depleted trijunction" they mention at the beginning of Sec. 3.

We agree with the referee that the full setup of our simulation was not clear. To address this point we added a panel to Figure 1 where we show the entire simulation domain and added a corresponding clarification in section 2.

> Another detail: How close to the center of the junction does the SC pairing Delta go? This could become important in the case with a phase difference.

This is explicitly stated in the text. Specifically, after Eq. (3) we write "The superconducting pairing is absent in the normal region". The "normal region" is defined above Eq. (1) by referencing Fig. 1 (c).

> (2) The derivation and motivation of Eq. 4 is a bit confusing. The authors admit this part to be "heuristic" - but they should probably explain (a) that the motivation is to extract the coupling matrix elements $\Gamma_{ij}$ (b) why we need $\Gamma$ to characterize the junction given that ultimately one could expect that the goal is to maintain a large gap at the trijunction and still couple the Majoranas at the far ends of the wire.

We agree with the referee that the motivation of the effective Hamiltonian derivation was not clear. Because the goal of optimizing the couplings $\Gamma_{ij}$ is to implement the braiding sequence, we have moved the braiding sequence to Fig. 1 and its corresponding explanation to the beginning of section 3.
Furthermore, we have also explained that we focus on the microcopic coupling of Majoranas closest to the middle region, and therefore we project out the far Majoranas.

> It would be good if Sec 3 could explain the context of Eq. 4 i.e. the need for defining the $\Gamma_{ij}$ as well as the motivation for the parameters $\delta_{+-}$ better.

We thank the referee for this suggestion. We have expanded the motivation of Eq. 4 and $\Gamma_{ij}$ at the beginning of section 3.
We believe that the role of $\delta_{+-}$ as a tool for the optimization procedure is clear, however, we have extended the explanation of its role in the loss function right before Eq. 6.

> For example should Eq 5 have absolute value functions on them? $\Gamma_{ij}$ cannot be positive, because they have to be anti-symmetric to maintain PHS.

We agree with the referee, and we have updated $delta_{+-}$ accordingly.

> (3) As apparent from the end of the last comment. Part of the issue is that the particle-hole symmetry of the effective Hamiltonian is not clear. The RHS of Eq. 4 is clearly particle-hole symmetric, assuming $\Gamma$ is an antisymmetric matrix. On the other hand, the LHS of Eq. 4 doesn't appear particle-hole symmetric unless one of $E_0,E_1, E_2$ is zero and the other two have the same magnitude and opposite signs.

We agree with the referee that the particle-hole symmetry of Eq. (4) is not clear and that the energies should be as described. In the new version we address this point by explicitly stating the particle-hole symmetry of the Hamiltonian, and the antisymmetry of the $\Gamma_{ij}$.

> There are a few other confusing aspects to the notation. The overlap matrix is defined in terms of $|\psi_j\rangle$, where this was not defined till this point. Assuming this is a typo and was meant to be $|\phi_j\rangle$, S appears to be a 3X3 matrix from Eq. 4.

We thank the referee for pointing out the confusing aspect of the notation. In this case $|\psi_j\rangle$ refers to a general eigenstate of three strongly coupled Majoranas which we decompose as a linear combination of three individual Majoranas. We have properly defined $|\psi_j\rangle$ referencing Fig 2 (b) where a coupled eigenstate is shown.

> On the other hand, the beginning of Sec 3 talks about the index j describing 6 states. One can make a reasonable guess based on the text that what is being talked about are the j corresponding to the three lowest states.

In the updated manuscript we explain that we study only the suitability of the trjunction for braiding, and Sec. 3 we state that we project away the eigenstates not localized in the trijunction.

> (4) Moving on the Sec 4, it is not clear that the state with "all three Majoranas coupled" can be described in terms of $\delta_{+-}$ i.e. it feels like this would have a very bad ratio of $\delta_+/\delta_-$.

We have clarified this aspect by renaming the sections and explaining that we use a different loss function that does not rely on $\delta_+$ or $\delta_-$ for tuning to the triple coupled point in Eq. 9.

> A minor notational issue is that the text refers to "left-top" and "right-top" which feels inconsistent with the R, L , T notation already introduced in Fig. 1a.

We have changed the notation to L, R and T everywhere.

> (5) In the sentence "linearly interpolates between the points where two and all Majorana states are coupled" in Sec. 4, it would be helpful to specify explicitly which state i.e. couplings are being referred to, maybe simply by referring to Fig 4c.

We now label each step of the braiding protocol, and in the new version we refer specifically to the steps between which we interpolate the voltages.

> (6) A somewhat technical point is that one imagines that conductance through a trijunction that leads to reasonable Majorana gaps might interfere with charging energy (first ingredient listed in Sec 4) because of screening through tunnel coupling as described in Phys. Rev. Lett. 122, 016801 (2019).

We agree with the referee. We have modified Sec. 3 to clarify that the competition between Coulomb and trijunction mediated couplings is beyond the scope of our work.

---

## Editorial Decision

published